# Primary prophylaxis with mTOR inhibitor enhances T cell effector function and prevents heart transplant rejection during talimogene laherparepvec therapy of squamous cell carcinoma

The application of mammalian target of rapamycin inhibition (mTORi) as primary prophylactic therapy to optimize T cell effector function while preserving allograft tolerance remains challenging. Here, we present a comprehensive two-step therapeutic approach in a male patient with metastatic cutaneous squamous cell carcinoma and heart transplantation followed with concomitant longitudinal analysis of systemic immunologic changes. In the first step, calcineurin inhibitor/ mycophenolic acid is replaced by the mTORi everolimus to achieve an improved effector T cell status with increased cytotoxic activity (perforin, granzyme), enhanced proliferation (Ki67) and upregulated activation markers (CD38, CD69). In the second step, talimogene laherparepvec (T-VEC) injection further enhances effector function by switching CD4 and CD8 cells from central memory to effector memory profiles, enhancing Th1 responses, and boosting cytotoxic and proliferative activities. In addition, cytokine release (IL-6, IL-18, sCD25, CCL-2, CCL-4) is enhanced and the frequency of circulating regulatory T cells is increased. Notably, no histologic signs of allograft rejection are observed in consecutive end-myocardial biopsies. These findings provide valuable insights into the dynamics of T cell activation and differentiation and suggest that timely initiation of mTORi-based primary prophylaxis may provide a dual benefit of revitalizing T cell function while maintaining allograft tolerance.

Locally advanced and metastatic cutaneous squamous cell carcinoma (cSCC) are more prevalent among immunosuppressed patients[1,2]. In addition, cSCC in immunosuppressed patients is less responsive to conventional treatment strategies than in immunocompetent patients[3]. Immune checkpoint inhibitors (ICIs) have emerged as a breakthrough treatment for advanced cSCC[4–6], but their application in solid organ transplant (SOT) patients is limited due to the potential risk of allograft rejection[7–9]. However, with a theoretically lower risk of graft rejection, intralesional immunotherapy offers a potentially safer alternative. Talimogene laherparepvec (T-VEC) is a genetically modified herpes simplex virus approved for treatment of locoregional advanced melanoma[10]. So far, none of these treatments have been formally tested in patients with SOT and an ongoing clinical trial is currently evaluating the efficacy of T-VEC in the treatment of cSCC

e-mail: michel.obeid@chuv.ch

(NCT03714828) where interim results showed a significant response with 100% complete response in stage 1 with a mean time to response as 43.4 days and the duration of the objective response rate as 190 days[11]. In addition, there are only rarely reported cases of T-VEC administration in heart[12] or in combined heart and kidney transplant patients[10] with inoperable recurrent melanoma. To date, only three cases of cSCC with T-VEC therapy in liver or kidney SOT patients have been previously reported[1,13,14]. In the first case study regarding a liver transplant patient with cSCC, immunosuppression was limited to a single treatment of low-dose mycophenolate mofetil (MMF) 250 mg twice daily without the concomitant use of calcineurin inhibitor (CNI) or corticosteroids (CS) and the patient achieved a complete and sustained remission after T-VEC therapy[1]. Importantly, the combination of ICI and CNI, a drug that causes T cell dysfunction by reducing T cell activation through lowering the translocation of nuclear factor of activated T cells (NFAT) into the nucleus, interleukin 2 (IL-2) production and IL-2 receptor expression[15], widely used for graft maintenance, has been associated with reduced anti-tumor response. In nine patients who continued to receive full-dose CNI during ICI therapy, there was no evidence of rejection, but only one of the nine patients had an objective tumor response[16,17].

Another important challenge in patients with prolonged CNI[18,19] treatment or chronic antigenic stimulation in the context of allograft tolerance[20] or cancer[21] is the presence of dysfunctional and/or exhausted T cells that are unresponsive to immunotherapeutic reprogramming and have historically been associated with poor antitumor responses. From a therapeutic perspective, T cell dysfunction is initially reversible, but may become severe and irreversible with time. In contrast to "anergic" states, early dysfunctional T cells can be reprogrammed whereas late dysfunctional T cells generated by chronic antigen stimulation and immunosuppressive microenvironment are refractory or less responsive to therapeutic reprogramming. Furthermore, chronic stimulation within a tumor/immunosuppressive microenvironment will facilitate the maintenance of tumor progenitor/non-dysfunctional CD8 + T cells that self-renew and the generation of more terminally differentiated dysfunctional T cells with the acquisition of a severe unprogrammed late dysfunctional state. As many patients with SOT fail to achieve a durable response to immunotherapy due to chronic T cell dysfunction or exhaustion, a key issue is to identify the most suitable intervention to ameliorate the hyporesponsiveness of SOT patients to immunotherapy. Several transplant patients experienced a tumor response without rejection when their immunosuppressive regimen contained mTORi sirolimus[16,22,23]. In contrast to CNI, mTORi offers multiple advantages such as anti-tumor activity[24] and preservation of Treg development[23] that is a key mediator of graft tolerance, which is strategically important because the use of PD-1/PD-L1 therapy was reported to impair Treg cells in the renal allograft[25]. Importantly, maintenance immunosuppression with the mTORi sirolimus and everolimus was reported to be associated with a significantly reduced risk of developing any post-transplant de novo malignancy and non-skin solid malignancies[26] and switching from CNI to mTORi had an antitumor effect in kidney transplant recipients with previous SCC[27]. Of note, advanced cSCC in SOT patients could be improved with adherence to annual dermatologic assessment, which mainly integrates early intervention, i.e. sunscreen awareness, lesion/field targeted therapy, early mTORi, reduced immunosuppression[28].

Recently, it has been reported that the addition of sirolimus as a secondary prophylaxis was associated with sustained anti-tumor efficacy while promoting allograft tolerance in melanoma patients with organ rejection and colitis induced by anti-PD-1 therapy[29].

The potential benefit of replacing calcineurin inhibitors (CNI) and mycophenolic acid (MPA) with mTORi as sole primary prophylaxis to prevent allograft rejection remains uncertain. In addition, the dynamic sequence of T cell activation and differentiation required to ameliorate dysfunctional T cell states, which is essential to achieve effective anti-tumor responses, is not fully understood.

In this study, we report a case of a SOT patient with cSCC treated with T-VEC with longitudinal monitoring of the immune system and analysis of key T cell markers such as cytotoxic activity, inhibitory marker expression, and differentiation profiles. We find that the timed immunomodulation with mTORi enhances T cell effector functions, increases Treg abundance and inflammatory cytokines while maintaining allograft tolerance under T-VEC therapy.

## Results
### Patient history
We followed a patient in his early sixties who underwent heart transplantation for ischemic heart disease several years prior to study entry and developed end-stage kidney disease (ESKD) requiring thrice-weekly hemodialysis. He received triple immunosuppression with MPA, tacrolimus (CNI) and low-dose daily prednisone. Allograft function was excellent in the first years after transplantation without rejection. After many years, he was diagnosed with metastatic cutaneous squamous cell carcinoma (cSCC). Despite multiple surgeries, his cSCC continued to progress. He developed multiple local lesions that became painful and ulcerated. A PET-CT revealed hypermetabolic cutaneous and subcutaneous lesions on the scalp, forehead, and temples, in addition to a hypermetabolic cervical lymph node with biopsy-proven squamous cell carcinoma metastasis (Fig. 1b, d). ICI and chemotherapy were not considered to be reasonable treatment options for him given his history of heart and kidney disease. We opted for intralesional immunotherapy with T-VEC, which is a potentially safer alternative with a theoretically lower risk of graft rejection (Fig. 1a). We stopped the dual immunosuppressants tacrolimus (CNI) and mycophenolate (MPA) and started an mTORi with everolimus prior to T-VEC injection. Thereafter, the patient remained exclusively on mTORi treatment without CNI and low-dose daily prednisone. Unfortunately, 18 days prior to T-VEC treatment, the patient contracted SARS-CoV-2 infection. He presented with mild symptoms (cough, rhinorrhea) and was started on sotrovimab the next day. COVID viremia decreased from 4.1 copies per milliliter (c/mL) to 2 c/mL within 15 days, with complete clinical recovery. The T-VEC injection was performed without any immediate complications and was limited to 1 million UFP/ml to allow for seroconversion due to the long duration of immunosuppression and the complexity of the clinical context. A few days later, he was admitted to the emergency room with fever (38.5 °C) and dyspnea. He was hospitalized for suspected E. coli pneumonia and treated with piperacillin/tazobactam for one week. Solumedrol was administered for one day, followed by rapid tapering of prednisone. Aphasia and dysarthria were noted 8 days after receiving T-VEC. Brain MRI showed no evidence of encephalitis or stroke. Due to the recent T-VEC injection, we performed PCR for HSV in blood and CSF: both were positive. In view of the T-VEC dissemination, the patient was treated with acyclovir for 10 days, with a rapid and complete recovery. 36 days after the T-VEC injection, the PET-CT showed a decrease in metabolic activity in the injected lesions, and the patient reported a decrease in pain in the necrotic treated area (Fig. 1c, e). T-VEC was not repeated due to the fragility of the patient. The patient did not experience any graft rejection or any immune-related adverse events. He was subsequently maintained on everolimus and low-dose daily prednisone and continued to receive dialysis three times a week.

**Early dysfunctional T cells states at baseline prior to mTORi treatment.** We first analyzed the absolute blood counts of CD4, CD8 T cells, B cells, NK cells and monocytes at baseline (V1), prior to mTORi treatment. Interestingly, we observed a severe T cell lymphopenia characterized mainly by decreased CD4 T cell absolute count at 125 cells/mm³ (normal reference interval: $N$ = 490−1640 cells/mm³)

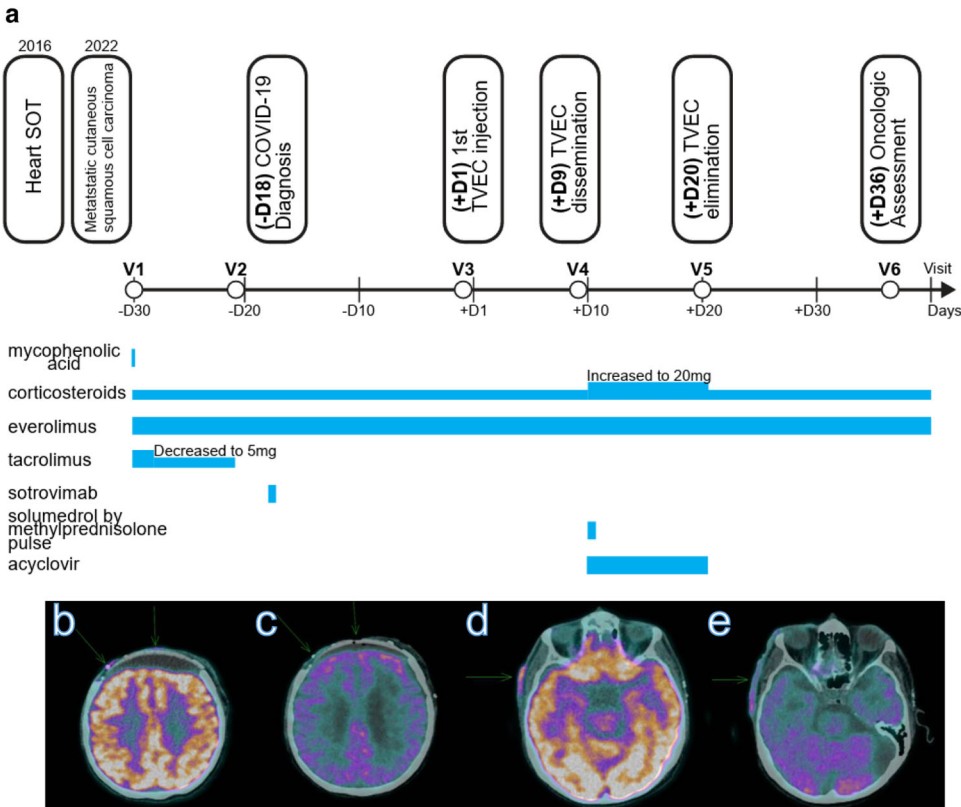

**Fig. 1 | Timeline and imaging of lesions. a** Patient clinical history and timeline for the establishment of mTOR and T-VEC injection. **b–e** PET-CT images showing cSCC lesions (pointed by green arrows) before T-VEC (**b**, **d**), decreasing 2 months after T-VEC injections (**c**, **e**).

and CD8 T cell absolute count at 79 cells/mm³ (normal reference interval: N = 170–880 cells/mm³) while natural killer cells (NK) were still within the normal reference interval (N = 80–690 cells/mm³) at 148 cells/mm³ (Fig. 2a). In addition, the CD4 Treg absolute counts were on the lower threshold (27 cells/mm³) compared to healthy patients (normal reference interval: N = 25–180 cells/mm³) (Fig. 2d) with increased expression of Bcl2, HLA-DR, TIGIT, and CD69. B cells (V1: 119 cells/mm³) and NK cells (V1: 148 cells/mm³) were within the established reference values of healthy controls at V1 whereas monocytes counts were above the upper threshold found in healthy patients (V1: 1110 cells/mm³, control upper limit: 800 cells/mm3) (Fig. 2a). In addition, EM and CM CD8 + T cells exhibited impaired cytotoxic effector function with low perforin and granzyme (GrzB) activity associated with low levels of inhibitory receptors (PD1 and TIM3) and activation markers (CD38, HLADR) but high levels of Bcl-2. This is compatible with a profile of early dysfunctional T cells that can be reprogrammed by immunotherapeutic intervention to evolve into more functional effector cells. The majority of CD4 and CD8 expressed CD38- Bcl2+ as well as CD38- GrzB- for CD8, indicating a resting state for T cells. T cell phenotype will be more detailed in the next paragraph.

**Improvement of effector T cells function after CNI and MPA interruption and mTORi treatment.** Following the dual stopping of CNI and MPA and the initiation of mTORi, in contrast to the persistent lymphopenia (Figs. 2a and 3a), we detected the beginning of the immune shift at V2 (9 days after V1) by the enhancement of the CD38+ Bcl2+ population and a switch from a Th2 phenotype to a Th1/Th17 and Th17 differentiation by V3 (20 days after V1) (Figs. 3b and 4a). We also observed a shift of CD4 CM to CD4 TM and EM and CD8 CM to CD8 EM at V3 (Fig. 3b, c). Interestingly, the total Treg counts remained below the reference threshold of healthy donors but the Treg frequencies within CD4 cells were increased well above the healthy controls at V2 and V3, indicating that increased Treg differentiation and/or survival

was occurring (Fig. 2b). HLA-DR and PD-1 increased at V2 in both memory CD4 and CD8 (Fig. 3d). All T helper (Th) subsets increased in the expression of Bcl2, CD38, ICOS, Ki67 and PD-1 from baseline V1 to V2 (Fig. 4b). In CD8, we observed a significant increase in the proportion of CD38- GrzB+ and a minor increase in both CD38+ GrzB- and CD38+ Bcl-2 + /− with a decrease in CD38- GrzB- in the EM population at V2 (Fig. 5a, b). Compared to healthy donors, we found increased expression of Bcl-2, CD25, HLA-DR and decreased expression of PD-1 at V1/V2 (Fig. 3e). Regarding the phenotypic profile, at V3, CD8 and CD4 CM were completely reduced while CD8 and CD4 EM were increased (Fig. 3c). Similarly, Treg frequencies were increased 1.5-fold by V2 compared to V1 (Fig. 3c). CD8 EM was also increased in CD38, Ki67, PD1, and TIGIT, however a slight reduction in Bcl-2 expression was observed, supporting a switch to more activated T cell profiles in both subsets (Fig. 5c). This immune state transition suggests a potential residual plasticity induced by mTOR initiation and CNI interruption, mainly manifested by an improvement of T cell effector function that could be more responsive to T-VEC than the initial dysfunctional states.

**Enhancement of effector function after T-VEC therapy.** T-VEC was administered 1 day after V3, but we began to observe a switch from CM to TM/EM profiles in both memory CD4 and CD8 from V3 along with a change from Th2 phenotype to Th17 and Th1/17 profiles in memory CD4 at V3. The effect of T-VEC administration was measured 8 days post-injection at V4. At this time, there were no significant changes in the total number of CD4 and CD8 T cells; however, significant effects were observed at both the phenotypic and functional levels. In CD4, the proportion of CD38+ Bcl2+ expressing cells increased in both CM and EM populations (Fig. 4a). Th1 differentiation was increased at V4 from V3, with increased expression of activation markers (CD69, CD25, CD38, ICOS, OX40), proliferation marker (Ki67), and co-inhibitory receptors (TIM3 and TIGIT) was observed compared to V3 (Fig. 4b). Th2 and Th17 subsets displayed similar activation profiles with

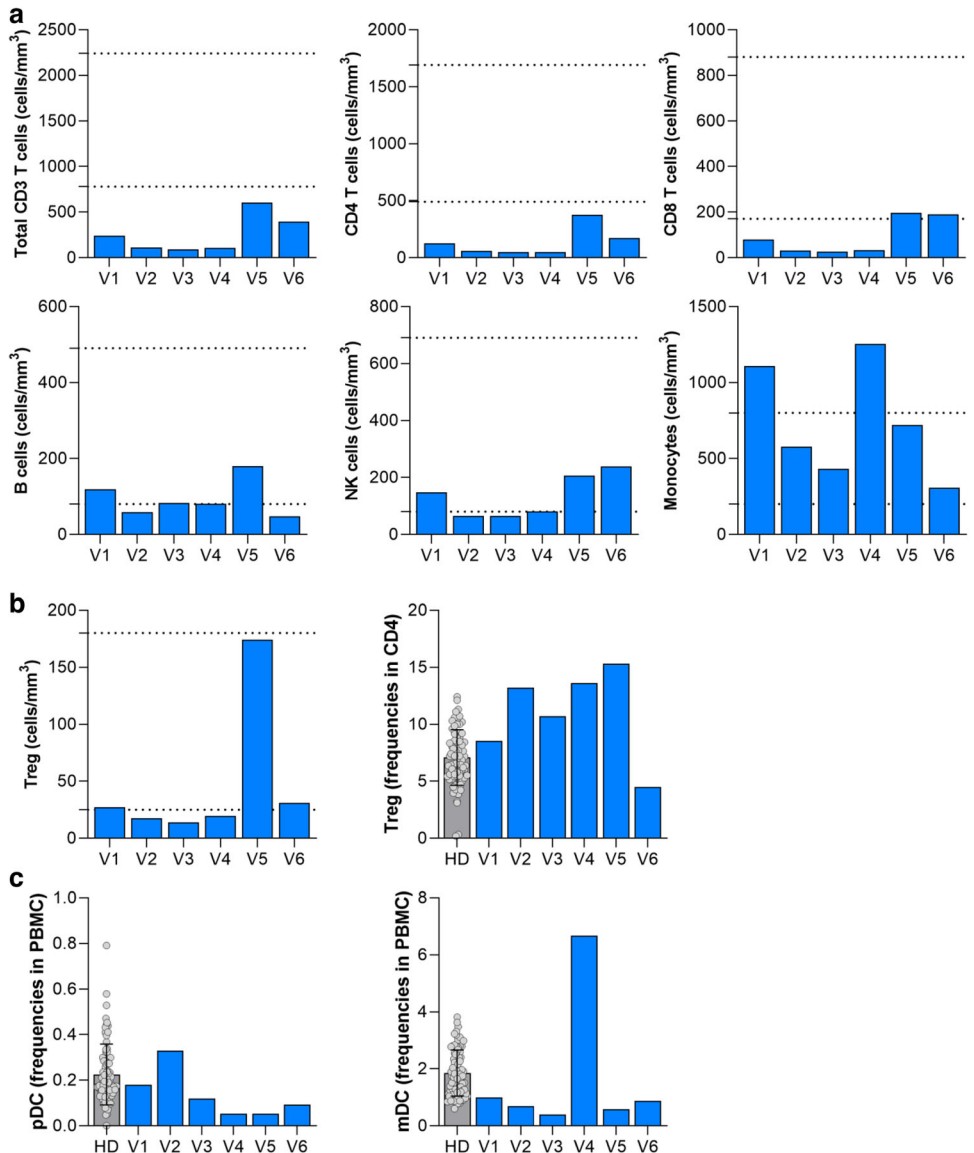

**Fig. 2 | Absolute counts and frequencies of immune cell subsets. a** Absolute counts of total CD3, CD4, CD8, B cells (CD19 + ), NK cells (CD56 + ), and monocytes (CD14 + ) over time. Dotted lines represent the minimum and maximum reference range from healthy donors (*N* = 450). **b** Absolute counts and frequencies of Tregs (T regulatory cell) (CD4 + CD45RO + CD127- CD25 + ) over time. Gray bar shown as mean ± SD of individual data points corresponds to reference frequencies from healthy donors (HD) (*N* = 74). **c** Frequencies of mDC (myeloid dendritic cells) (Lin- CD14- HLA-DR+ CD11c + ) and pDC (plasmacytoid dendritic cells) (Lin- CD14- HLA-DR + CD123 + ) over time. Gray bar shown as mean ± SD of individual data points corresponds to reference frequencies from HD (*N* = 74).

increased expression of CD69, CD38, ICOS, Ki67, and PD-1 from V3 (Fig. 4b). The effects post injection at V4 for CD8 were substantial. In CD8 CM, a large proportion of CD38+ GrzB- and CD38+ Bcl2+ cells were increased compared to V3 (Fig. 5a, b). An increase in CD38, GrzB, perforin, and PD-1 was observed in CD8 CM from V3 to V4 (Fig. 5c). Notably, Ki67, TIGIT and LAG3 decreased in CD8 CM and TIM3 expression was not observed (Fig. 5c). For CD8 EM at V4, an increase in the proportion of CD38+ GrzB +/− and CD38+ Bcl2+ expressing cells was observed (Fig. 3b, c). For the EM activation profile, Bcl-2, CD69, GrzB, perforin, PD1, and LAG3 were increased at V4 compared to V3. Interestingly, a significant increase in the total number of monocytes and the frequency of myeloid dendritic cells (mDC) were observed at V4 (Fig. 2a, c). The cytokine profile was measured at V4, showing a major switch to a more pro-inflammatory profile with increased BAFF, BNDF, EGF, HGF, IL-18, IL-1Ra, IL-6, IP10, CCL2 (MCP1), CCL3 (MIP-α), CXCL10, CCL4 (MIP-1β), PDGF, CCL5, VEGFA, and sCD25 (Fig. 3a). No significant changes were observed in the total number of B and NK cells at V4 (Fig. 2a).

During T-VEC dissemination, the highest number of both total CD4, CD8, and Tregs were observed at V5, 20 days after T-VEC administration (Fig. 2a, b). At V5, CD4 CM showed nearly equal proportions of CD38- Bcl2 + , CD38+ Bcl2+ and CD38+ Bcl2- expressing cells and for CD4 EM, approximately 50% of the cells were CD38+ Bcl2- phenotype (Fig. 4a). Memory CD4 maintained a TM/EM phenotype and differentiated to the Th1 subset. However, activation markers in Th1 were similarly expressed from V4 apart from increased CD57 expression (Fig. 4b). The remaining Th2 and Th17 CD4 again expressed a similar activation profile from V4 except for increases in CD69, CD25, CD57, ICOS, TIM-3, TIGIT, and OX40 in Th2 and increases in ICOS, Ki67, and TIGIT in Th17 (Fig. 4b). For CD8 T cells at V5, CD8 CM appeared to decrease in CD38+ GrzB- but increase in CD38+ Bcl2- cells (Fig. 5a, b). For activation, Ki67 and PD-1 levels remained the same from V4 but increases in CD69, CD25, and TIGIT and decreases in GrzB, perforin, and Bcl2 were observed (Fig. 5c). On the other hand, CD8 EM showed a substantial increase in the proportion of CD38+ GrzB-, CD38+ GrzB + , and CD38+ Bcl2- T cells (Fig. 5a, b). From V4, increases in CD25, CD38,

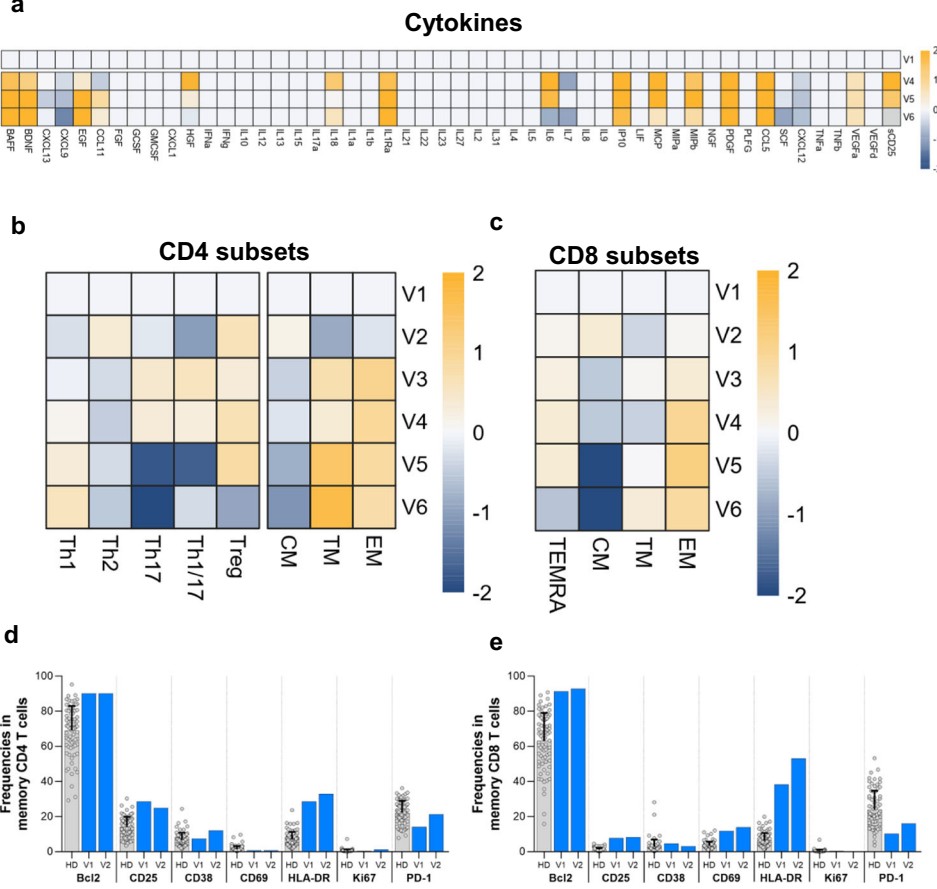

**Fig. 3 | Cytokine and T cell differentiation profiles. a** Differential cytokine expression from baseline compared to post T-VEC injection represented as a heatmap and expressed as log2-fold change compared to V1. Increasing expression represented in orange gradient, no changes shown as white, and decreasing expression shown in blue gradient compared to V1. **b** Differentiation of CD4 subsets in memory CD4 (T helper cell (Th), T regulatory cell (Treg), central memory (CM), transitional memory (TM), effector memory (EM)) over time represented as a heatmap and expressed as log2-fold change compared to V1. Increasing expression represented in orange gradient, no changes shown as white, and decreasing expression shown in blue gradient compared to V1. **c** Differentiation of CD8 subsets in memory CD8 (central memory (CM), transitional memory (TM), effector memory (EM), T effector memory CD45RA + (TEMRA)) over time represented as a heatmap and expressed as log2-fold change compared to V1. Increasing expression represented in orange gradient, no changes shown as white, and decreasing expression shown in blue gradient compared to V1. **d, e** Expression of activation markers at V1 and V2 vs HD in memory CD4 (**d**) and in memory CD8 (**e**). Gray bar shown as mean ± SD of individual data points corresponds to reference frequencies from HD (*N* = 74).

HLA-DR, Ki67, PD-1, TIM-3, and OX40 expression and decreases in Bcl2, CD69, CD57, perforin, TIGIT, LAG3, and NKG2D expression were observed, indicating a more differentiated, exhausted phenotype (Fig. 5c).

The detected majority decrease in cytotoxic activity (GrzB) within CD38 + CD8 combined with the heightened expression of inhibitory markers (PD1, TIM3), indicates a secondary exhaustion reprogramming of T cells, resembling a more differentiated and dysfunctional state akin to terminally differentiated T cells. This transformation seems to be driven by the elevated inflammatory state associated with the widespread dissemination of T-VEC. This emerging cell shift seems to be triggered by the heightened inflammatory state resulting from the dissemination of T-VEC. In addition, a shift in B cell phenotype was observed at V5; the highest total number of B cells was observed, and NK cells were also increased compared to V4 (Fig. 2a). Interestingly, monocytes and mDCs decreased significantly compared to V4, indicating the cessation of the innate inflammatory phase (Fig. 2a). By V5, the proinflammatory cytokine profile had decreased with reduced IL-6, IL-18, HGF and sCD25, while the anti-inflammatory cytokine ILRA increased (Fig. 3a).

**The return to a less activated T cells states upon antiviral treatment.** After elimination of T-VEC, a reduction in immune activation was

evaluated at V6. Absolute counts of total CD4, CD8 and Tregs were shown to be significantly reduced at V6 (Fig. 2a, b). CD4 TM/EM were maintained with a significantly reduced CM population (Fig. 3b). CD4 subsets differentiated towards Th1 starting at V4 and maintained a highly skewed Th1 profile by V6. The activation profile for all three Th subsets was dramatically reduced to or below baseline levels at V5 compared to V1, except for OX40 expression for Th1 and elevated PD-1 and Ki67 expression for all three subsets (Fig. 4b). In CD8, the proportion of cells expressing CD38- GrzB- returned to similar levels as at baseline; however, a small fraction of CD38+ cells remained in both CD8 CM and EM while a greater proportion of Bcl-2- was present in CD8 CM and EM compared to baseline (Fig. 5a, b). The CD8 TM/EM profile was also similar to the CD4 memory subsets (Fig. 5c). Strikingly, in the activation profile, the CD8 EM subset lost expression of activation markers compared to baseline except for CD38, Ki67, and NKG2D, while the CD8 CM subset decreased expression of CD69, CD25, LAG3, TIGIT, and NKG2D, but increased expression of CD38, CD57, GrzB, perforin, and PD-1 compared to baseline (Fig. 5c). At V6, absolute B cell and monocyte counts were reduced to below baseline levels, and DC subset frequencies were reduced to below baseline levels at V6 (Fig. 2a, c). However, NK cell counts were increased at V6 compared to V5 (Fig. 2a). The cytokine expression profile was similar in the types of cytokines produced from V5, but with notable decreases in CCL11,

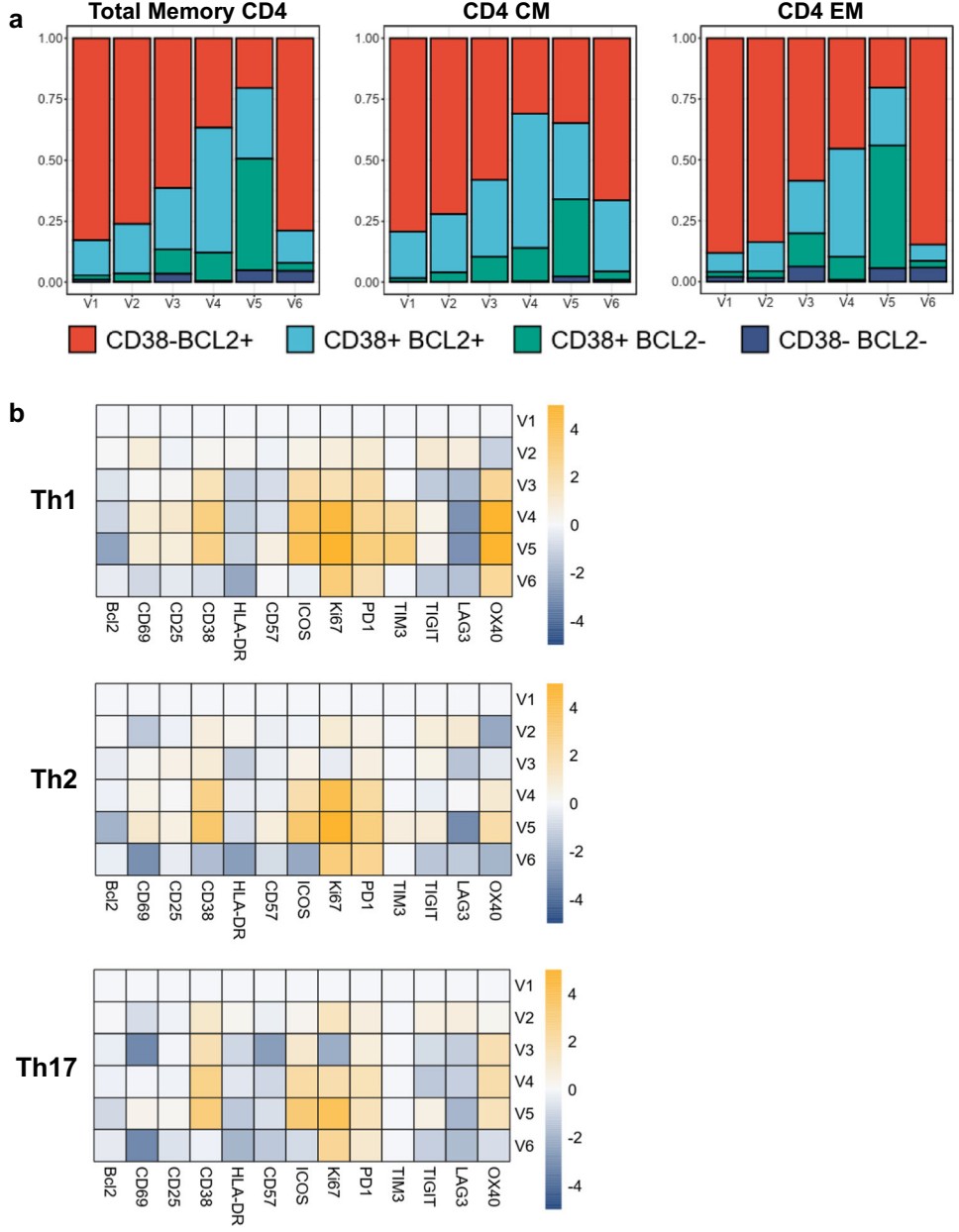

**Fig. 4 | CD4 activation. a** Proportion of CD38/ Bcl-2 expression in CD4 memory, central memory (CM), and effector memory (EM) over time shown as stacked barplots. Red bar represents CD38- Bcl2 +, light blue bar represents CD38+ Bcl2 +, green bar represents CD38+ Bcl2-, dark blue bar represents CD38- Bcl2-. **b** Activation and co-inhibitory profile of Th1, Th2, and Th17 subsets in memory CD4 over time represented as a heatmap and expressed as log2-fold change compared to V1. Increasing expression represented in orange gradient, no changes shown as white, and decreasing expression shown in blue gradient compared to V1.

HGF, IL-6, IL-7, CCL2 (MCP1), CCL4 (MIP-1β), SCF and slight increases in CXCL13 and IL-18 (Fig. 3a).

## Discussion

Due to concerns regarding alloimmunity, organ rejection and an increased risk of developing de novo cancer after transplantation, solid organ transplant (SOT) patients have generally been excluded from cancer immunotherapy clinical trials[30,31]. Under cancer immunotherapy, high allograft rejection rates ranging from 30 to 40% have been reported in many systematic reviews and institutional experiences[8,9]. In addition, despite being rare (0.09%), severe myocarditis caused by cancer immunotherapy in non-transplant patients can be associated with high mortality rates ranging from 36% to 67%[32]. Due to the risk of both rejection and myocarditis, studies in heart transplant recipients treated with cancer immunotherapy are limited and should be conducted with extreme caution. Therefore, the use of

primary prophylaxis may be an appropriate strategy to mitigate the risk of rejection. Despite T-VEC dissemination and CNI and MPA interruption, our patient had no signs of allograft rejection under mTORi primary prophylaxis. mTORi treatment induced an important Treg[23] development from naïve T cells[34], which is potentially central to enhancing graft tolerance. Mechanistically, it has been reported that AKT decreases TGF-β-induced Foxp3 expression in a kinase-dependent manner and through a rapamycin-sensitive pathway, and that the expression of active AKT selectively impairs CD4+Foxp3+ differentiation[35]. Conversely, it is well established that CNI but not mTORi, decreases proportions of Treg cells in transplant recipients[36]. Using the FOXP3 reporter mouse model, rapamycin has been shown to promote de novo (TGFβ-dependent) switching of alloantigen-specific CD4 + T cells to Treg cells whereas CNI abrogates this process[37], which is a major advantage of mTORi over CNI in this specific setting. Interestingly, in contrast to the blockade of both signal 1 and signal 2 of

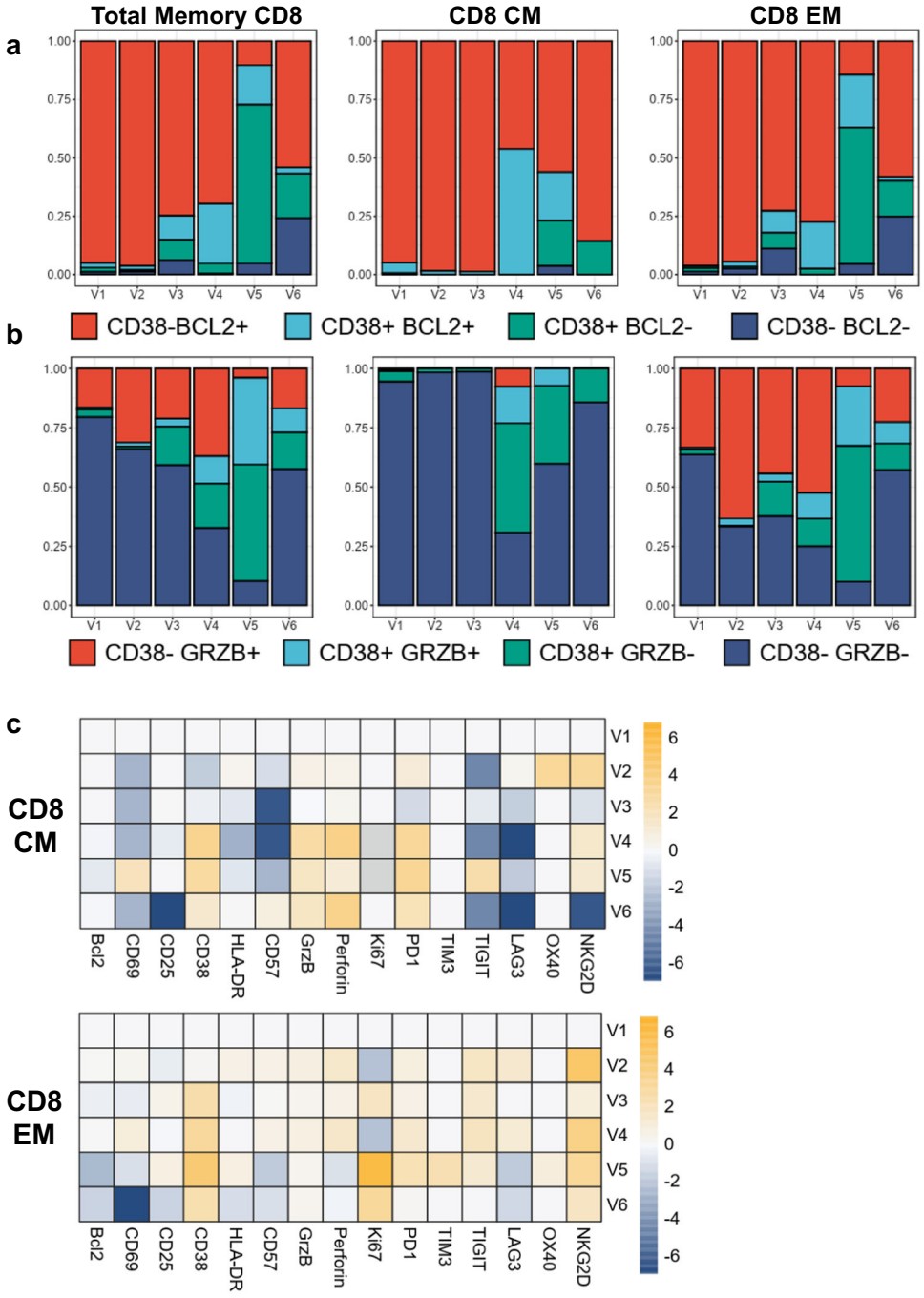

**Fig. 5 | CD8 activation. a**, **b**, Proportion of CD38/ GrzB expression (**a**) and CD38/ Bcl-2 expression (**b**) in CD8 memory, central memory (CM), and effector memory (EM) over time shown as stacked barplots. For (**a**), red bar represents CD38- Bcl2 + , light blue bar represents CD38+ Bcl2 + , green bar represents CD38+ Bcl2-, dark blue bar represents CD38- Bcl2- and for (**b**), red bar represents CD38- GrzB + , light blue bar represents CD38+ GrzB + , green bar represents CD38+ GrzB-, dark blue bar represents CD38- GrzB-. **c,** Activation and co-inhibitory profile of CD8 CM and EM subsets represented as a heatmap and expressed as log2-fold change compared to V1. Increasing expression represented in orange gradient, no changes shown as white, and decreasing expression shown in blue gradient compared to V1.

T cell activation by CNI, concomitant treatment with mTORi and immune co-stimulation resulted in massive apoptosis of alloreactive T cells and allograft tolerance[38]. Collectively, CNI interruption and mTORi led to the induction of a more allograft tolerant immune situation through the significant increase in Treg and the decrease in alloreactive T cells, which may potentially act at multiple regulatory levels to prevent allograft rejection, myocarditis, and other organ toxicities. As far as we know, this case report is the first study to describe the success of mTORi as primary prophylaxis when administered in a time-efficient manner concurrent with CNI interruption prior to cancer immunotherapy. The successful use of mTORi as

secondary prophylaxis to prevent a second renal allograft rejection during ICI[29] has been reported previously. In this case, the diagnosis of renal rejection is not certain because the patient presented with multiple concomitant immune-related adverse events (irAEs) and no renal biopsy was performed, keeping the differential diagnosis of ICI-related nephritis instead of allograft rejection possible, especially since the other concomitant irAEs respond well and disappear during immunosuppressive therapy.

Allowing an antitumor response is another prerequisite for the implementation of primary prophylaxis in the context of cancer[39]. Interestingly, despite the single and reduced dose of TVEC, we

observed a rapid morpho-metabolic response in the treated lesions. In addition, the patient had a clinical response with less pain and significantly reduced ulceration at the injected sites, and the PET-CT showed a reduction in activity. From a mechanistic perspective, we observed a shift in the profile of memory T cells toward more effector states with cytotoxic and proliferative activities shortly after treatment with mTOR inhibitors and interruption of CNIs. This shift was characterized by an increased frequency of activated Th1-polarized T cells. Most probably, T cell remodeling was observed as a result of the simultaneous cessation of MPA and the introduction of mTORi at V1. We were unable to exclusively compare the individual effects of the immunosuppressive interventions at any specific point in time. However, V1 served as the baseline immunological profile for MPA + CNI, V2 represented the early effects of tapered CNI + mTORi, and the remaining visits studied the effects of mTORi only. Between V1 and V2 (Fig. 2a), the frequency of T cell subsets did not change dramatically; however, in Fig. 3b, c, we observed a phenotypic shift toward Th2/Tregs in memory CD4, CM in memory CD4/8, and TEMRA/EM in memory CD8. In vivo models for cSCC have demonstrated increased infiltration of CM/EM under clinically relevant doses of rapamycin[40]. Additionally, previous studies have shown that regulatory T cells (Tregs) are selectively maintained[41] as we observed by increased frequencies following administration of the mTOR inhibitor (mTORi). The levels of PD-1 increased between V1 and V2 in response to T cell activation, which was demonstrated by increased expression of HLA-DR.

Additional studies identified a variety of immunostimulatory effects of mTORi on a wide range of immune cells at different stages in the development of both the innate and adaptive immune responses[42]. This includes an intriguing immune-enhancing effect on memory T cell proliferation[43]. More specifically, mTORC1 were reported to activate glycolysis and lipid biosynthesis to support effector T cell differentiation[44]. mTORC are now known to promote Th1, Th2, and Th17 differentiation. Accumulating evidence suggests that mTORi may promote favorable immune modulation at multiple levels by synergizing with tumor vaccines[45] or immune checkpoint inhibitors[46–48]. In addition, the addition of mTORi to ICI reduced the frequency of exhausted TILs (tumor-infiltrating lymphocytes) while increasing the frequency of activated Th1-polarized T cells in the tumor. Moreover, a Th1-polarized proinflammatory cytokine profile was promoted by primary innate immune cells[47]. mTORi was reported to promote effector T cell survival, which was associated with the accumulation of CD25$^{hi}$ Bcl-2$^{hi}$ cells in culture and the accumulation of CD8 + T effector cells within the tumor[47]. mTORi was demonstrated to reduce T cell exhaustion in patients with bladder cancer. This may suggest that mTORi could reverse T cell exhaustion[49]. In our patient, we found that mTORi ameliorated the dysfunctional T cell states by increasing cytotoxic and proliferative activities concomitantly with an increase in Bcl-2 expression in activated CD8 + T cells and CD4+ effector cells as well as in Treg cells. Interestingly, mTORi is known to be a promoter of autophagy[50]. During immune responses to chronic viral infections, autophagy has been reported to enhance effector T cell survival and differentiation[51]. In contrast, the use of CNI markedly reduced PD-1 expression through their effect on NFATc, which is likely to reduce the ability of exhausted T cells to be reinvorgated[52]. The immunostimulatory effects of mTORi are not limited to CD8 + T cells, as mTORi also potentiate the cytotoxic effects of γδ T cells, thereby enhancing their antitumor efficacy[53]. Therefore, while mTORC1 inhibition promotes CD8+ memory T cell differentiation, it also reduces CD8+ memory T cell function and CD8+ effector T cell expansion[54]. Therefore, determining the dose and schedule of mTORi required to stimulate the proliferation of CD8+ memory T cells without compromising the expansion of the effector cells will be an important consideration in cancer therapy.

T-VEC has been shown to induce immunogenic cell death in melanoma cell lines with associated release of damage-associated molecular patterns (DAMPs), including release of high-mobility group box-1 (HMGB-1), adenosine triphosphate (ATP), and ecto-calreticulin (CRT)[55] along with other oncolytic properties such as the upregulation of tumor-derived antigens in an immunostimulatory microenvironment, local production of GM-CSF, and cross-priming of CD8 + T cell responses by dendritic cells that facilitate an anti-tumor immune response[56]. Various combinations of systemic immunotherapeutic interventions, including ICIs, have been proposed to enhance the efficacy of T-VEC[57]. mTORi, which provides additional immune activation, may be a potentially attractive adjuvant strategy to deplete depleted TILs and enhance activated Th1-polarized T cells[47].

To our knowledge, the specific effects of T-VEC treatment on the immune response, in particular the shift from CM to EM T cells combined with Th1 polarization, have not been reported in the context of clinical trials or patient data. In fact, an in vitro stimulation model described that T-VEC increased the release of pro-inflammatory cytokines such as IL-2 by activated T cells and IL-6 by mDC[58]. It is plausible that T-VEC dissemination may be a potential co-inducer of this systemic immune stimulation. However, comprehensive longitudinal studies investigating the specific effects of T-VEC on EM T cells and Th1 polarization are currently lacking. Further research is needed to fully understand the multifaceted interactions between T-VEC and the immune system, particularly with regard to its immunostimulatory effects on EM T cells and the Th1 immune response.

Although the infection was at the end stage with a significantly reduced viral load, we cannot formally exclude that the SARS-CoV-2 infection did not act as a second immunostimulant. Indeed, the development of SARS-CoV-2-specific T cells has been reported during infection but the phenotype of the SARS-CoV-2-specific T cell memory response was largely CD4[59,60], with most CD4+ cells expressing a CM profile[61]. In addition, it was reported that all CD4 T cell populations, CD8 T cells, total, naive and EM cell populations were significantly decreased compared to healthy donors in ICU and non-ICU patients, while only CM CD8 T cells were significantly increased and TEMRA was unchanged[62]. In contrast, the immunostimulatory effect in our patient was different as we observed a major shift from CM to EM in both CD8 and CD4 with increase in TEMRA accompanied by a major shift to a Th1 profile with an increase in cytotoxic activity. Regarding PD-1 upregulation, increased Fas and PD-1 expression in both CD4+ and CD8 + T cells has been reported in patients with SARS-CoV-2 infection[63] and an enhancement of exhausted PD-1 expressing T cells in critically ill SARS-CoV-2 patients[64]. While in our patient we observed an increase in inhibitor markers (PD-1, TIM3) prior to SARS-CoV-2 infection at the time of mTOR induction. Since T-VEC is known to secrete GM-CSF, which plays a role in stimulating the differentiation and activation of monocytes and dendritic cells, it is important to recognize that SARS-CoV-2 infection itself can also induce immunostimulatory effects on these immune cells[65].

Importantly, T-VEC binds to cell surface receptors such as nectin-1 and herpesvirus entry mediator A, which are widely expressed on a variety of human cell types[66]. Despite the fact that T-VEC is an attenuated form of HSV-1 that has been modified to reduce viral pathogenicity, patients with compromised immunity may be at high risk for life-threatening systemic viral replication, thereby warranting cautious use including multidisciplinary consideration and careful monitoring and supervision by an experienced team.

There are several limitations to this study that must be acknowledged. This finding was restricted to one subject and requires further confirmation with a larger population of SOT patients receiving T-VEC. Our research revealed an increase in memory T cell activation and differentiation resulting from mTORi modulation and T-VEC administration. However, no functional experiments were conducted to examine the T cell response's specificity and magnitude in greater detail. Further investigation is necessary due to the limited understanding of T cell responses during T-VEC therapy. Previous research

has indicated higher densities of tumor-infiltrating memory T cells at the lesion site following localized T-VEC administration. The combination of T-VEC and anti-PD-1 therapy has also shown to improve responses[67,68]. Despite an increase in clonal CTLs at the injected and non-injected lesion sites between 1-5 weeks post T-VEC injection, a recent study found no detection of HSV-1-specific TCR clonotypes[69]. Further studies are needed to fully characterize the T cell response regarding TCR specificity/clonality, as well as tumor antigens involved and potential HSV-1-specific responses.

Finally, dysfunctional T cells, which often result from prolonged CNI and MPA treatment, can be reprogrammed and revitalized by timely initiation of mTORi-based primary prophylaxis.

These reinvigorated T cells can potentially lead to improved antitumor efficacy of T-VEC and potentially other type of cancer immunotherapies. In addition to previously reported benefits from studies that have suggested lower risk of certain malignancies, lower mortality, fewer cardiac allograft vasculopathy-related events, and lower risk of renal dysfunction with the use of mTORi[70], this dual benefit of mTORi may hold significant promise in the context of cancer immunotherapy for SOT recipients. By unraveling the underlying mechanisms involved in T cell exhaustion, differentiation, and response to immunotherapy, we can better optimize treatment strategies and potentially improve patient outcomes. Advancements in our understanding of these complex immune processes will pave the way for personalized therapeutic approaches tailored to individual patients, leading to more effective management of metastatic cancers in the context of organ transplantation.

This study suggests that the timely use of mTORi represents a step in the right direction to achieve better cancer treatment outcomes while preserving the integrity and function of the transplanted organ. However, it is essential to conduct further research and clinical trials to validate these findings and to optimize the use of mTORi-based primary prophylaxis in this specific population.

## Methods

### Sample collection and ethics approval
Our research complies with all relevant ethical regulations approved by the CHUV Ethics Committee. The patient's blood was collected at each visit and processed for the mass cytometry (CyTOF) and cytokines analysis. Patient provided informed consent. Serum and blood samples for normal reference values were collected from 450 healthy individuals to establish reference ranges for absolute counts of blood cell populations and concentrations of serum immune signatures. Blood samples from 74 healthy individuals were used to establish reference values for T cell phenotype and activation profiles. Healthy individuals provided informed consent.

### Immune profiling of blood immune cell populations by mass cytometry
Patient blood was processed following a standardized whole blood staining protocol[71]. 200 μL of blood containing immune cells were incubated for 30 min at room temperature (RT) with a 50 μL antibody cocktail of metal-conjugated antibodies against CD3, CD7, CD45, CCR4, CCR6, CCR7, CXCR3, CXCR5, CD127, and TCR γδ (Standard BioTools). Cells were washed with PBS (Laboratorium Dr. G. Bichsel AG) and fixed with 2.4% PFA (Thermo Fisher Scientific) for 10 min at room temperature (RT), lysed for 15 min at RT using 4 mL of Bulklysis solution (Cytognos), and then Bulklysis solution washed off. Following another PBS wash, cells were incubated for 30 min at RT with metal-conjugated antibodies against CD141, CD69, CD8, CD4, IgA2, CD19, ICOS, IgG3, CD31, IgD, IgA1, IgG1, CD123, CD21, CD62L, CD3, CD27, CD10, CD14, CD1c, CD11c, CD45RO, CD24, CD38, CD66b, CD25, CD45RA, CD20, IgM, TCRαβ, HLA-DR, PD1, CD56, IgG2, and CD16. Cells are then washed and total cells were identified by DNA intercalation (1 μM Cell-ID Intercalator, Standard BioTools) in 2% PFA at 4 °C

overnight. Labeled samples were acquired using the HELIOS CyTOF system (Standard BioTools) and FCS files were normalized to EQ Four Element Calibration Beads using the CyTOF software. The complete CyTOF panel for immune populations is shown in Supplementary Table 1 and gating strategy detailed in Supplementary Fig. 1.

### Immune Profiling of T cell phenotype, activation, and inhibition
Patient blood was processed following a standardized whole blood staining protocol[71]. 200 μL of blood containing immune cells were incubated for 30 min at 4 °C with a 50 μL antibody cocktail of metal-conjugated antibodies against CD8, CD4, CCR4, CD127, CCR6, CXCR3, CCR7, CXCR5, and CD45 (Standard BioTools). Cells were washed with PBS (Laboratorium Dr. G. Bichsel AG) and fixed with 2.4% PFA (Thermo Fisher Scientific) for 10 min at room temperature (RT), lysed for 15 min at room temperature using 4 mL of Bulklysis solution (Cytognos), and then Bulklysis solution washed off. Following another PBS wash, cells were then incubated for 30 min at RT with a 50 μL antibody cocktail of metal-conjugated monoclonal antibodies directed against CD3, CD19, ICOS, TIGIT, OX40, PD1, CD95, CD62L, CD27, CD25, CD45RO, NKG2D, CD38, CD66b, TIM-3, CD45RA, LAG3, HLA-DR, CD56, CD57, CD71, and CD16. Cells were then permeabilized for 30 min at 4 °C using 1 mL of Fix/Permeabiliztion buffer from the Foxp3 Fixation/Permeabilization Kit (eBioscience), then washed with Permeabilization Wash buffer and stained for 30 min at 4 °C with a 50 μL antibody cocktail of metal-conjugated monoclonal antibodies against Ki67, Bcl2, perforin, and granzyme B. Cells were then washed and total cells were identified by DNA intercalation (1 μM Cell-ID Intercalator, Standard BioTools) in 2% PFA at 4 °C overnight. Labeled samples were acquired using the HELIOS CyTOF system (Standard BioTools) and FCS files were normalized to EQ Four Element Calibration Beads using the CyTOF software. Complete CyTOF panel for T cell phenotype is shown in Supplementary Table 1.

### Profiling of serum immune signatures
Serum concentrations of cytokines (IL-1α, IL-1β, IL-6, IL-12p70, TNF-α, TNF-β, IFN-γ, IFN-α2, IL-2, IL-7, IL-15, IL-4, IL-5, IL-8, IL-9, IL-10, IL-13, IL-17A, IL-18, IL-21, IL-22, IL-23, IL-27, IL-31, and BAFF), cytokine receptor IL-1RA, soluble CD25, chemokines (CCL3, CCL4, CCL5, CCL11, CXCL1, CXCL8, CXCL9, CXCL10, CXCL12 and CXCL13), and growth factors (BDNF, PDGF, PLGF, VEGFa, VEGFd, EGF, NGF-β, FGF-2, HGF, LIF, SCF, GM-CSF and G-CSF) were determined by Luminex ProcartaPlex (Thermo Fisher Scientific)[72] immunoassays for each marker and reference values were determined by 450 sera collected from healthy individuals.

### Reporting summary
Further information on research design is available in the Nature Portfolio Reporting Summary linked to this article.

## Data availability
Data used in the preparation of this manuscript are available within the Article, Supplementary Information, Supplementary Data and Source Data file. There are no restrictions on data access. Raw data files from mass cytometry and luminex are available at https://doi.org/10.5281/zenodo.10412950 from the corresponding authors on request. Further information and requests for resources and reagents should be directed to the corresponding authors. Source data are provided with this paper.

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

## Acknowledgements

We thank Laura Molinari of the Centre d'immunothérapie et de vaccinologie for providing us with the samples required to conduct this study, the implementation of experimental protocols, and for technical assistance. We thank Pr Olivier Michielin for participating in the medical discussion. We would also like to thank Michael Moulin, Manon Geiser, and Craig Fenwick for managing and processing the CyTOF samples. We would finally like to thank the patient for their selfless participation. We are grateful to Robin Bartolini for assistance with the graphic abstract. The graphical abstract was created with Biorender.com.

## Author contributions

M.O. conceived the study and research design. M.O., K.A., O.G., S.L., R.H., R.D., M.P. collected clinical data and provided clinical care. F.M., D.D. contributed to the curation of the data. V.J. performed the mass cytometry analysis and prepared the figures. V.J., A.N., F.M., D.D., S.P., G.P., and M.O. contributed to the data analysis and data visualization. M.O. supervised the study and drafted the paper. All authors contributed to the writing of the manuscript and all authors approved the final manuscript.

## Competing interests

All authors declare no competing interests.

## Additional information

Victor Joo[1,6], Karim Abdelhamid [2,6], Alessandra Noto[1], Sofiya Latifyan[2], Federica Martina[1], Douglas Daoudlarian [1],
Rita De Micheli[2], Menno Pruijm[3], Solange Peters [2], Roger Hullin[4], Olivier Gaide [5], Giuseppe Pantaleo [1] &
Michel Obeid [1] ✉

[1]Centre Hospitalier Universitaire Vaudois (CHUV), University of Lausanne, Department of Medicine, Immunology and Allergy Division, Rue du Bugnon 46, CH-
1011 Lausanne, Switzerland. [2]Centre Hospitalier Universitaire Vaudois (CHUV), University of Lausanne, Oncology Department, Rue du Bugnon 46, CH-1011
Lausanne, Switzerland. [3]Centre Hospitalier Universitaire Vaudois (CHUV), University of Lausanne, Department of Medicine, Nephrology Division, Rue du
Bugnon 17, CH-1011 Lausanne, Switzerland. [4]Centre Hospitalier Universitaire Vaudois (CHUV), University of Lausanne, Cardiology, Cardiovascular Depart-
ment, Rue du Bugnon 46, CH-1011 Lausanne, Switzerland. [5]Centre Hospitalier Universitaire Vaudois (CHUV), University of Lausanne, Dermatology Division,
Rue du Bugnon 46, CH-1011 Lausanne, Switzerland. [6]These authors contributed equally: Victor Joo, Karim Abdelhamid. ✉e-mail: michel.obeid@chuv.ch

