## [Peer Review File · Nature Communications]

Primary prophylaxis with mTOR inhibitor enhances T-cell effector function and prevents heart transplant rejection during talimogene laherparepvec therapy of squamous cell carcinomaREVIEWER COMMENTS

Reviewer #1 (Remarks to the Author):

In organ transplant patients, the combination of long-term immunosuppression together with the carcinogenic properties of some immunosuppressants together with the "normal" risk factors lead to an increase in the quantitative and qualitative risk of skin cancer and especially cSCC with aggressive biological behavior compared to the general population. On the other hand, the treatment options are limited, especially the immunotherapy that caused a revolution in the treatment of inoperable cSCC, increases the risk of allograft rejection, therefore there is a clinical need for effective and safe treatment options for cSCC in this complex population.

This case report presents a cardiac transplant recipient with inoperable metastatic cSCC treated by switching his previous immunosuppression to mTOR inhibitors and subsequently adding T-VEC with therapeutic success (at least initially). The clinical process is accompanied by an immunological "dissection" showing the result of each clinical stage. The combination between the clinic and the laboratory data is fascinating.

This case adds to only two former published cases reporting the safe administration of T-VEC in heart transplants. The beneficial effect of the treatment in this patient together with the basic immune process, provide essential information and may offer a relatively safe treatment option for these complex patients.

I have the following remarks:

1. In line 118 describing the patient's history: " In the year 2020, he was diagnosed with metastatic cutaneous squamous cell carcinoma. Due to multiple local progression of his cSCC, he underwent many several surgeries..."

Please provide more details about what other therapeutic strategies were applied? i.e., primary prevention, lesion/field directed therapy, acitretin etc. ? Was he followed regularly in a dermatology clinic for OTR?

2. The patient received T-VEC only once with initial partial tumor regression, since then you have at least one year of follow up please provide the current clinical details.

3. There is abundant published data regarding mTORi in cancer prevention, and specifically secondary prevention of cSCC in OTR (Kauffman HM et al. Transplantation 2005;80:883-89 and more). Moreover, the importance of early introduction of mTORi therapy in OTR (Euvrard S et al. NEJM 2012; 367:4 and more..) could be nicely explained by "the fact that dysfunctional T cells, which often result from prolonged CNI and MPA treatment, can be reprogrammed and revitalized by initiation of mTORi-based immunosuppression ," therefore it should be introduced into your discussion

4. Since V-TEC is currently FDA approved for melanoma it is important to add that current trials are underway determining the efficacy of T-VEC to treat CSCC (NCT03714828).

5. Finally, you should mention the advanced cSCC in OTR could be at least in part prevented by early intervention i.e., sun protection awareness, lesion/field directed therapy, early mTORi introduction, lower levels of immunosuppression etc. (Chan, A-W, etc. Am J Transplant. 2019; 19: 522-531)

Minor comments

In line 1 please add before mTOR- mammalian target of rapamycin

Make sure the writing is uniform, for example:
change VTEC to V-TEC (lines: 28, 64,67,138,139,190, 192, 195, 217, 218, 248,etc.)

mTOR inhibition or mTORi whatever you choose but it has to be uniform (for example in line: 311 mTORi inline 315: mTOR inhibitors)

Once you define (line 45) central memory (CM) you use (CM) for the rest of the text, the same for effector memory (EM) (line 46)

In conclusion: this is an important well written case report which sheds light on the underlying immunological process following different clinical measures and provides hope for additional therapy for OTR with unresectable cSCC. The text is clear, the figures have corresponding legends, and with the suggested references regarding mTORi the reference list will appropriately cover previous literature.

Reviewer #2 (Remarks to the Author):

This manuscript by Joo et al. reports changes in immune cell numbers and phenotypes across time in a single cardiac transplant recipient with cutaneous squamous cell carcinoma. This is a major clinical dilemma in current oncologic practice. In this patient the clinical investigators transitioned the patient from calcineurin inhibitor immunosuppressive agent to the mTOR inhibitor, everolimus and also used T-VEC, an oncolytic virus to treat the patients squamous cell carcinoma. The authors reports that there was an improvement in the number and cytotoxic phenotype of effector T cells following transition to mTORi therapy while maintaining a Treg population that is likely necessary for allograft tolerance. Further, they showed that T-VEC treatment was generally well tolerated and associated with a change in CD4+ and CD8+ T cells from a central memory to an effector memory phenotype. The importance of this work is both in the focus on an important unmet medical need, and in providing a more comprehensive analysis of immune cell changes in this population. The work also provides a potentially innovative roadmap for further clinical development in solid organ transplant patients who develop advanced cutaneous malignancies. The work will likely be highly cited, but I did have some minor issues for the authors to consider, as detailed below.

1. The complexity of the patient's clinical course is not surprising, however some of the specific details were confusion. For example, did the patient only receive a single dose of T-VEC? T-VEC treatment usually starts with a lower dose to allow seroconversion and then uses a higher dose so it may be worth noting the dosing and number of injections received.
2. Further, it was not clear to me if the patient responded to the limited T-VEC treatment or not. Please clarify the tumor response and whether the patient had any further cancer therapy, and if not, how long the patient has been followed with some clinical response. The photos in Fig. 1 f-h were not very easy to interpret but appreciate this cannot be changed.
3. The changes in T cell phenotype observed with the transition to everolimus are a very important observation. However, is the T cell remodeling due to the mTORi or due to removing the mycophenolic acid? Did this patient remain on mTORi or go back on to another calcineurin inhibitor or other immunosuppressive agent(s) after T-VEC treatment?
4. The effector memory emergence upon exposure to T-VEC is consistent with prior reports. Have the authors done any work to understand the antigen specificity of the T cell response? It would be interesting to note whether the T cell changes are viral-specific or may also be tumor-specific.
5. The authors should include some comment in the Discussion on the limitations of this study, including the single report as well as the lack of antigen specificity and functional analysis of the immune cells.

POINT-BY-POINT REPLY TO REFEREE NO. 1

We thank the reviewer for carefully reading the manuscript and providing us with their constructive comments.

The reviewer summarized our study favorably and positively, concluding that:

“The combination between the clinic and the laboratory data is fascinating. This case adds to only two former published cases reporting the safe administration of T-VEC in heart transplants. The beneficial effect of the treatment in this patient together with the basic immune process, provide essential information and may offer a relatively safe treatment option for these complex patients. This is an important well written case report which sheds light on the underlying immunological process following different clinical measures and provides hope for additional therapy for OTR with unresectable cSCC. The text is clear, the figures have corresponding legends, and with the suggested references regarding mTORi the reference list will appropriately cover previous literature.”

However, the reviewer notes several comments.

*1. In line 118 describing the patient's history: " In the year 2020, he was diagnosed with metastatic cutaneous squamous cell carcinoma. Due to multiple local progression of his cSCC, he underwent many several surgeries..."
Please provide more details about what other therapeutic strategies were applied? i.e., primary prevention, lesion/field directed therapy, acitretin etc. ? Was he followed regularly in a dermatology clinic for OTR?*

Response:

The patient had a very regular dermatologic follow-up with primary prevention care and initially several local resections since June 2020. He then underwent resection surgery by a team of neurosurgeon and otolaryngologist for skull infiltration. Then, each time a new symptomatic cutaneous or subcutaneous lesion appeared; a new resection was performed by his dermatologist. Finally, he was referred to us to discuss systemic therapy due to multisite progression.

2. The patient received T-VEC only once with initial partial tumor regression, since then you have at least one year of follow up please provide the current clinical details.

Response

The first T-VEC injection was limited to 1 million UFP/ml to allow for seroconversion due to the long duration of immunosuppression and the complexity of the clinical context. After this first injection, the patient had a metabolic oncologic response in all injection sites with necrotic lesions. This was also corroborated by a significant increase in herpetic serology compared to the pre-TVEC baseline. Therefore, our therapeutic strategy was to treat the

remaining metastatic sites. Unfortunately, the patient decided to discontinue treatment due to weariness from the long years of treatment and was admitted to palliative care 3 months later.

It is important to note that despite the single and reduced dose of TVEC, we observed a rapid morpho-metabolic response in the treated lesions. In addition, the patient had a clinical response with less pain and significantly reduced ulceration at the injected sites, and the PET-CT showed a reduction in activity, the images were not taken on the same machine, so it is not as obvious as one would expect.

The text of the patient's history and discussion has been updated (lines 139, 145 and 323).

3. There is abundant published data regarding M-TORi in cancer prevention, and specifically secondary prevention of cSCC in OTR (Kauffman HM et al. Transplantation 2005;80:883–89 and more). Moreover, the importance of early introduction of M-TORi therapy in OTR (Euvrard S et al. NEJM 2012; 367:4 and more..) could be nicely explained by "the fact that dysfunctional T cells, which often result from prolonged CNI and MPA treatment, can be reprogrammed and revitalized by initiation of mTORi-based immunosuppression," therefore it should be introduced into your discussion

Response

We appreciate the reviewer's comment, which we believe succinctly explains our reasoning. We have included it in the introduction., line 99

4. Since V-TEC is currently FDA approved for melanoma it is important to add that current trials are underway determining the efficacy of T-VEC to treat CSCC (NCT03714828).

Response

Thank you for this excellent suggestion. We have added information regarding that specific clinical trial (NCT03714828) to highlight ongoing efforts in utilizing T-VEC for CSCC in the introduction, Line 66.

5. Finally, you should mention the advanced cSCC in OTR could be at least in part prevented by early intervention i.e., sun protection awareness, lesion/field directed therapy, early mTORi introduction, lower levels of immunosuppression etc. (Chan, A-W, etc. Am J Transplant. 2019; 19: 522–531)

Response

Thank you for this excellent suggestion. We have added this information about preventative measures for the development of cSCC in SOT recipients (introduction Line 105).

Minor comments

In line 1 please add before mTOR- **mammalian target of rapamycin (done)**

Make sure the writing is uniform, for example:

change VTEC to **V-TEC** (lines: 28, 64,67,138,139,190, 192, 195, 217, 218, 248,etc.) **(done)**

mTOR inhibition or **mTORi** whatever you choose but it has to be uniform (for example in line: 311 mTORi inline 315: mTOR inhibitors) **(done)**

Once you define (line 45) central memory (CM) you use (CM) for **the rest of the text**, the same for effector memory (EM) (line 46) **(done)**

In conclusion: this is an important well written case report which sheds light on the underlying immunological process following different clinical measures and provides hope for additional therapy for OTR with unresectable cSCC. The text is clear, the figures have corresponding legends, and with the suggested references regarding mTORi the reference list will appropriately cover previous literature.

POINT-BY-POINT REPLY TO REFEREE NO. 2

We thank the reviewer for carefully reading the manuscript and providing us with their constructive comments.

The reviewer summarized our study favorably and positively, concluding that:

“This manuscript by Joo et al. reports changes in immune cell numbers and phenotypes across time in a single cardiac transplant recipient with cutaneous squamous cell carcinoma. This is a major clinical dilemma in current oncologic practice. In this patient the clinical investigators transitioned the patient from calcineurin inhibitor immunosuppressive agent to the mTOR inhibitor, everolimus and also used T-VEC, an oncolytic virus to treat the patients squamous cell carcinoma. The authors reports that there was an improvement in the number and cytotoxic phenotype of effector T cells following transition to mTORi therapy while maintaining a Treg population that is likely necessary for allograft tolerance. Further, they showed that T-VEC treatment was generally well tolerated and associated with a change in CD4+ and CD8+ T cells from a central memory to an effector memory phenotype. The importance of this work is both in the focus on an important unmet medical need, and in providing a more comprehensive analysis of immune cell changes in this population. The work also provides a potentially innovative roadmap for further clinical development in solid organ transplant patients who develop advanced cutaneous malignancies. The work will likely be highly cited, but I did have some minor issues for the authors to consider, as detailed below.”

However, the reviewer notes several comments.

1. *The complexity of the patient’s clinical course is not surprising, however some of the*

specific details were confusion. For example, did the patient only receive a single dose of T-VEC? T-VEC treatment usually starts with a lower dose to allow seroconversion and then uses a higher dose so it may be worth noting the dosing and number of injections received.

Response

The first T-VEC injection was limited to 1 million UFP/ml to allow for seroconversion due to the long duration of immunosuppression and the complexity of the clinical context. After this first injection, the patient had a metabolic oncologic response in all injection sites with necrotic lesions. This was also corroborated by a significant increase in herpetic serology compared to the pre-TVEC baseline. Therefore, our therapeutic strategy was to treat the remaining metastatic sites. Unfortunately, the patient decided to discontinue treatment due to weariness from the long years of treatment and was admitted to palliative care 3 months later. The text of the patient's history has been updated (lines 139, 145).

2. Further, it was not clear to me if the patient responded to the limited T-VEC treatment or not. Please clarify the tumor response and whether the patient had any further cancer therapy, and if not, how long the patient has been followed with some clinical response. The photos in Fig. 1 f-h were not very easy to interpret but appreciate this cannot be changed.

Response

Thank you for this excellent question.

It is important to note that despite the single and reduced dose of TVEC, we observed a rapid morpho-metabolic response in the treated lesions. In addition, the patient had a clinical response with less pain and significantly reduced ulceration at the injected sites, and the PET-CT showed a reduction in activity, the images were not taken on the same machine, so it is not as obvious as one would expect.

The text of the discussion has been updated (lines 323).

3. The changes in T cell phenotype observed with the transition to everolimus are a very important observation. However, is the T cell remodeling due to the mTORi or due to removing the mycophenolic acid? Did this patient remain on mTORi or go back on to another calcineurin inhibitor or other immunosuppressive agent(s) after T-VEC treatment?

Response

Thank you for this excellent question.

Most probably, T cell remodeling was observed as a result of the simultaneous cessation of MPA and the introduction of mTORi at V1. We were unable to exclusively compare the individual effects of the immunosuppressive interventions at any specific point in time. However, V1 served as the baseline immunological profile for MPA+CNI, V2 represented the early effects of tapered CNI + mTORi, and the remaining visits studied the effects of mTORi only.

Between V1 and V2 (Figure 2A), the frequency of T cell subsets did not change dramatically; however, in Figure 3B/C, we observed a phenotypic shift toward Th2/Tregs in memory CD4, CM in memory CD4/8, and TEMRA/EM in memory CD8. In vivo models for cSCC have demonstrated increased infiltration of CM/EM under clinically relevant doses of rapamycin [30228949]. Additionally, previous studies have shown that regulatory T cells (Tregs) are selectively maintained, as we observed by increased frequencies following administration of the mTOR inhibitor (mTORi) [17182569]. The levels of PD-1 increased between V1 and V2 in response to T cell activation, which was demonstrated by increased expression of HLA-DR.

The patient remained exclusively on mTORi treatment without CNI and low-dose daily prednisone.

We have added this important information into the Discussion on line 330.

4. The effector memory emergence upon exposure to T-VEC is consistent with prior reports. Have the authors done any work to understand the antigen specificity of the T cell response? It would be interesting to note whether the T cell changes are viral-specific or may also be tumor-specific.

Response

Thank you for this excellent question.

We concur with the reviewer that investigating the antigen specificity of both tumor and viral antigens, as well as the clonal diversity of the T cell response, would have provided exciting data. However, our study was unable to include these investigations during the patient's treatment course. Consequently, we have acknowledged these limitations in the Discussion section on line # 422.

5. The authors should include some comment in the Discussion on the limitations of this study, including the single report as well as the lack of antigen specificity and functional analysis of the immune cells.

Response

Yes, we also agree with the reviewer that it is important to discuss the limitations of the study. We have now added in the Discussion on line 422 highlighting the single case study design, the absence of functional assays, and the need for studying antigen-specific responses as mentioned above in Question 4.

REVIEWERS' COMMENTS

Reviewer #1 (Remarks to the Author):

I think the revised version of the manuscript has improved and is now worthy of publication. This will add important data to the growing body of evidence regarding T-VEC in OTR. This case report has also a unique and interesting immunological "dissection" showing the outcome of each clinical stage.

Unfortunately, I noted two additional comments:

1. In line 71 , To date, only one case of cSCC has been previously reported....

In fact, I found at least two other case reports that were published recently:

a. Miller M et al, case rep Dermatol 2023 Jan-Dec; 15(1): 99-104

b. Lebhar J et al, JAAD case rep. 2023; 40: 53-57

2. In line 119-120 "Due to multiple local progression of his cSCC, he underwent many several surgeries..."

Please change to: he underwent many surgeries or he underwent several surgeries

Reviewer #2 (Remarks to the Author):

The authors have responded to my concerns adequately.

Reviewer #1 (Remarks to the Author):

I think the revised version of the manuscript has improved and is now worthy of publication. This will add important data to the growing body of evidence regarding T-VEC in OTR. This case report has also a unique and interesting immunological "dissection" showing the outcome of each clinical stage.

Unfortunately, I noted two additional comments:

1. In line 71 , To date, only one case of cSCC has been previously reported....

In fact, I found at least two other case reports that were published recently:

a. Miller M et al, case rep Dermatol 2023 Jan-Dec; 15(1): 99–104

b. Lebhar J et al, JAAD case rep. 2023; 40: 53–57

2. In line 119-120 "Due to multiple local progression of his cSCC, he underwent many several surgeries..."

Please change to: he underwent many surgeries or he underwent several surgeries

We thank the reviewer for their time and for finding relevant references.

We have added these two additional case reports and changed the paragraph accordingly as follows:

1. To date, only three cases of cSCC with T-VEC therapy in liver or kidney SOT patients have been previously reported. [32897942, 37711513, 37383323]

2. We also changed the redundancy for "many several..." to "several..."